# Plasma methionine metabolic profile is associated with longevity in mammals

N. Mota-Martorell [1], M. Jové[1], R. Berdún[1] & R. Pamplona [1✉]

Methionine metabolism arises as a key target to elucidate the molecular adaptations underlying animal longevity due to the negative association between longevity and methionine content. The present study follows a comparative approach to analyse plasma methionine metabolic profile using a LC-MS/MS platform from 11 mammalian species with a longevity ranging from 3.5 to 120 years. Our findings demonstrate the existence of a species-specific plasma profile for methionine metabolism associated with longevity characterised by: i) reduced methionine, cystathionine and choline; ii) increased non-polar amino acids; iii) reduced succinate and malate; and iv) increased carnitine. Our results support the existence of plasma longevity features that might respond to an optimised energetic metabolism and intracellular structures found in long-lived species.

[1] Department of Experimental Medicine, Lleida University-Lleida Biomedical Research Institute (UdL-IRBLleida), Lleida, Catalonia, Spain.
✉email: reinald.pamplona@mex.udl.cat

All living organisms use the same 20 amino acids for protein synthesis. However, it seems that the minimum number to build functional proteins is around ten[1]. Indeed, the first proteins were built with the available amino acids generated upon the early earth via abiotic processes[2]. This abiotic set probably comprised the amino acids alanine (A), aspartate (D), glutamate (E), glycine (G), isoleucine (I), leucine (L), proline (P), serine (S), threonine (T) and valine (V)[2]. These early amino acids were encoded adaptively by the early genetic code. The evolutionary transition from this initial genetic code to a genetic code where all 20 amino acids are fixated seems to be linked to the adaptations demanded by the appearance of oxygen in the biosphere during early evolution[3,4]. Thus, amino acids with increased redox properties, such as arginine (R), asparagine (N), cysteine (C), glutamine (Q), histidine (H), lysine (K), methionine (M), phenylalanine (F), tyrosine (Y) and tryptophan (W) were fixated into the new genetic code as adaption to preserve aerobic life[3].

Interestingly, the protein compositional content of the sulfur amino acids methionine and cysteine is species-specific and is associated with animal longevity. Thus, long-lived animal species show the lower methionine[5–7,8] and cysteine[9] protein content, surely as adaptive response[4,10] to the low rate of endogenous damage and highly resistant macromolecular components also present in longevous species[11,12,10]. Reinforcing these observations, the free tissue methionine content is also lower in diverse long-lived animal species[13,14]; and the pro-longevity effects of nutritional (methionine restriction, MetR)[15,11] and pharmacological (metformin)[16] interventions are mediated by changes in methionine metabolism. In addition to its role in several intracellular processes, methionine is the core of a complex metabolic network which can be divided in three parts: methionine cycle, the transsulfuration pathway, and

polyamine biosynthesis[17,18] (for details see Fig. 1). Significantly, manipulation of each of these branches affects longevity in diverse experimental animal models[15]. Consequently, available findings point to the metabolism of methionine as a key target to study the molecular adaptive mechanisms underlying differences in animal longevity.

The purpose of this study was to investigate the methionine metabolic phenotype of long-lived mammalian species. Specifically, we have designed a study to detect and quantify a panel of metabolites including 35 different molecular species in plasma of eleven mammalian species showing more than one order of magnitude of difference in longevity, from 3.5 years in mice to 120 years in humans. The metabolites detected and quantified were: (1) methionine and its related metabolites, including the intermediates of the transmethylation pathway S-adenosylmethionine (SAM), S-adenosylhomocysteine (SAH) and homocysteine; betaine as metabolites involved in the regeneration of methionine plasma levels; the intermediates of the transsulfuration pathway cysteine and cystathionine; taurine and glutathione as downstream metabolites of the transsulfuration pathway; and vitamin B6 metabolites pyridoxal and pyridoxamine, as cofactors of the transsulfuration enzymes; (2) additional amino acids including eight non-polar amino acids (alanine, glycine, leucine, isoleucine, phenylalanine, proline, tryptophan and valine), four polar uncharged amino acids (asparagine, serine, threonine and tyrosine), two polar negatively charged amino acid (aspartate and glutamate) and two polar positively charged amino acids (arginine and histidine); (3) TCA cycle metabolites, including pyruvate, citrate, α-ketoglutarate, succinate, fumarate and malate; and (4) methionine-derived lipid intermediates, such as choline and carnitine. The plasma metabolites profile was determined using a LC-MS/MS platform to define specific phenotypic profiles associated with animal longevity.

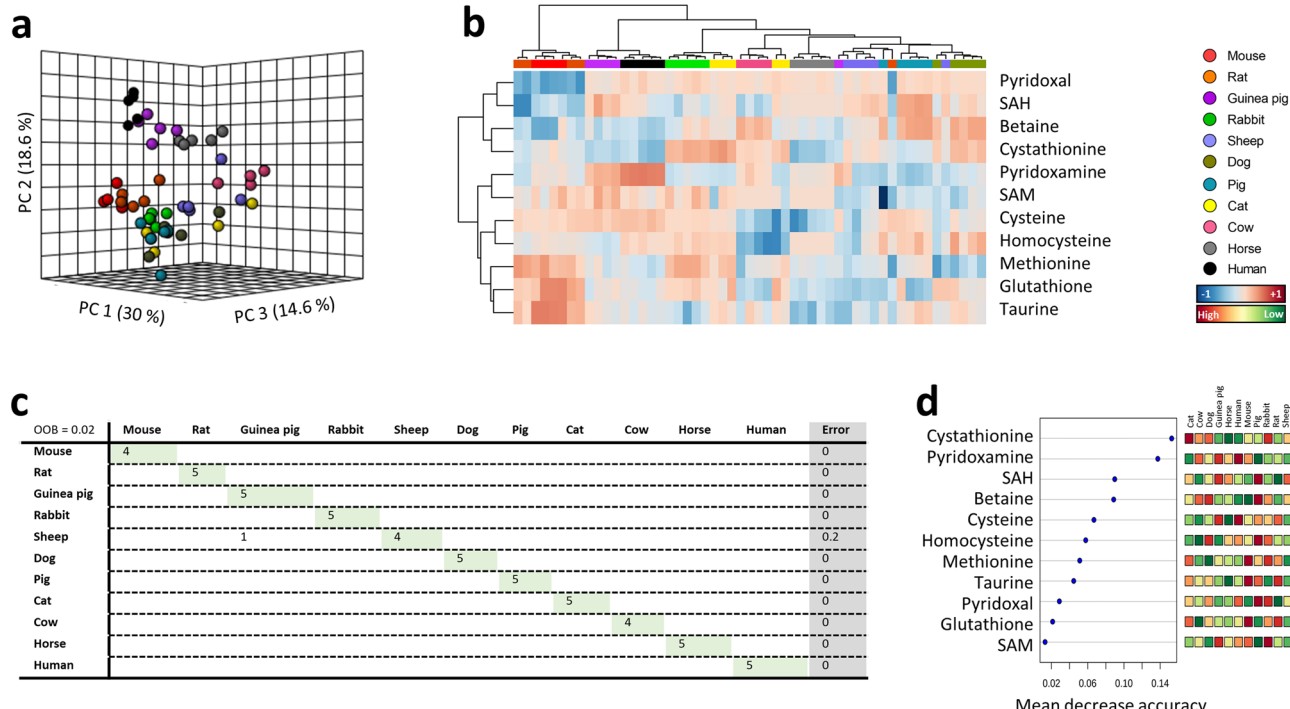

**Fig. 1 Multivariate statistics reveals a species-specific methionine plasma profile. a** Principal component analyses (PCA) representation of methionine-related metabolites. X: Principal component 1 (PC1); Y: Principal component 2 (PC2); Z: Principal component 3 (PC3). **b** Hierarchical clustering of individual animal samples according to metabolite abundance. **c** Random forest (RF) classification algorithm. **d** VIP scores for RF. Data values are obtained from four (mice and cows), five (rats, rabbits, guinea pigs, sheep, dogs, pigs, cats and horses) or six (humans) specimens.

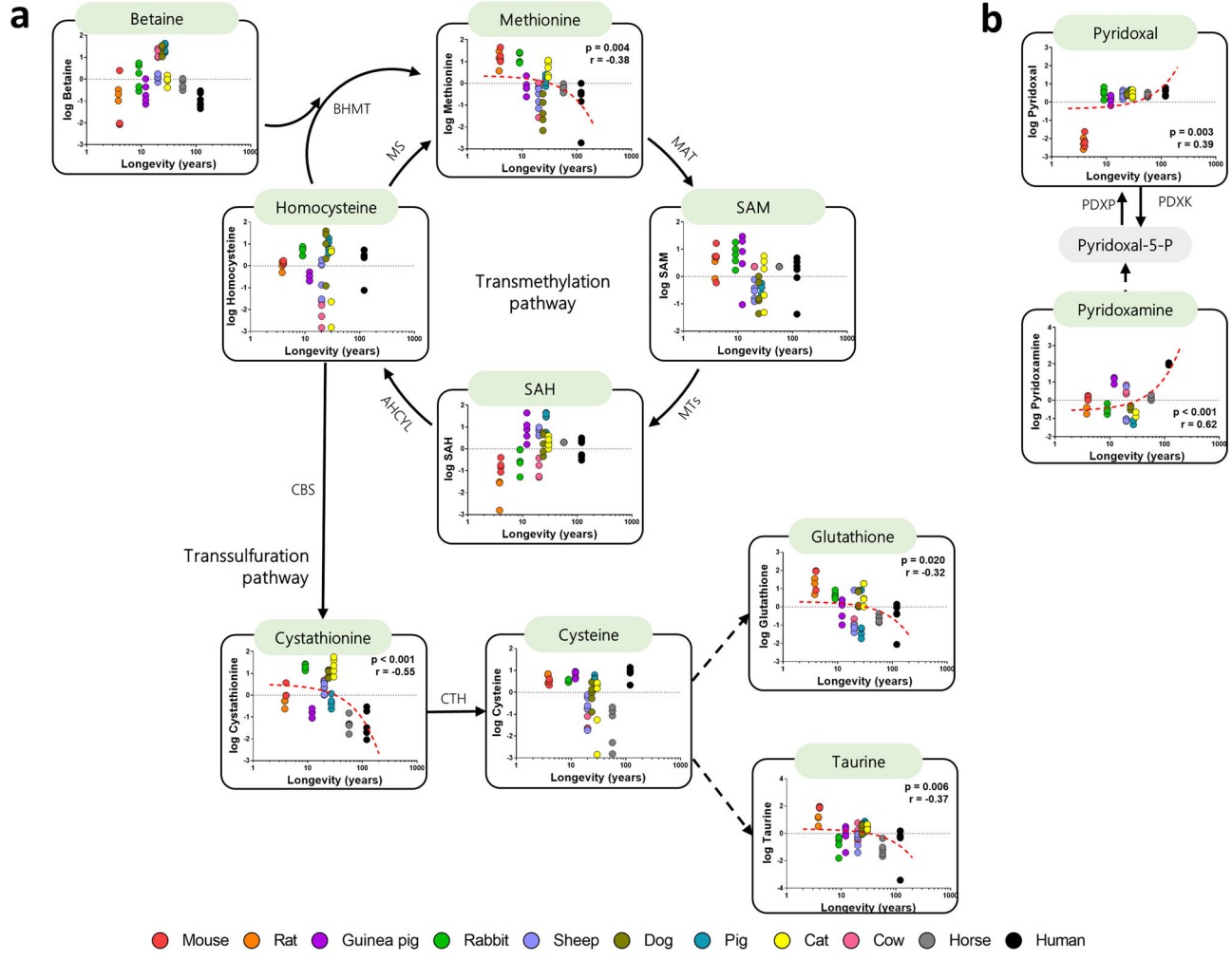

**Fig. 2 Plasma methionine-related metabolites and its correlation with animal longevity.** Individual plasma correlation of the metabolites with animal longevity involved in the transmethylation and transsulfuration pathways (**a**) and vitamin B6 metabolites (**b**). Dashed lines refer to reactions in which more than one enzyme is involved. Grey-shaded boxes refer to non-detected metabolites. Pearson correlation was performed. Minimum signification level was set at $p < 0.05$. All metabolites were log-transformed in order to accomplish the assumptions of normality. Data values are obtained from four (mice and cows), five (rats, rabbits, guinea pigs, sheep, dogs, pigs, cats and horses) or six (humans) specimens. Enzyme codes refer to: MS methionine synthase, BHMT betaine-homocysteine S-methyltransferase, MAT methionine adenosyltransferase, MTs methyltransferases, AHCYL adenosylhomocysteinase-like 1, CBS cystathionine-β-synthase, CTH cystathionine-γ-lyase, PDXK pyridoxal kinase, PDXP pyridoxal phosphatase.

## Results

**Multivariate statistics reveals a species-specific methionine-related metabolites plasma profile.** In order to determine whether plasma methionine and its related metabolites concentration differed among mammals, multivariate statistics were applied. Non-supervised principal component analysis (PCA) suggested the existence of a species-specific methionine-related metabolites plasma profile (Fig. 1a), capable to explain 63.2% of sample variability. A hierarchical clustering of the samples represented by a heat map revealed a good sample clusterization according to animal species (Fig. 1b). Accordingly, the performance of a Random forest (RF) classification algorithm revealed a species overall classification error <0.1 (Fig. 1c), being cystathionine and pyridoxamine the variables with the highest contribution to classification accuracy (Fig. 1d). These data suggest that RF is a good model to identify specimens of specific mammalian species according to its plasma methionine metabolism profile.

**Methionine and cystathionine plasma content are decreased in long-lived animals.** The specific changes in plasma methionine and its related metabolites content across animal longevity was

evaluated (Fig. 2a, Supplementary Tables 2 and 3). Specifically, methionine plasma content was decreased in long-lived animals, as well as the sulphur-containing metabolites cystathionine, taurine and glutathione. However, transmethylation metabolites, such as SAM, SAH and betaine, as well as the sulphur-containing metabolites homocysteine and cysteine, remained unchanged across animal longevity. Vitamin B6 intermediate metabolites pyridoxal and pyridoxamine, in turn, were increased in long-lived animals (Fig. 2b).

**Plasma amino acid profile predicts animal species.** Since plasma methionine is associated to animal longevity, we hypothesized the possibility that other amino acids could also be involved in the achievement of increased longevity. Specifically, we have been able to unambiguously detect 16 additional amino acids apart from cysteine and methionine. In order to determine whether plasma amino acid content defines animal species, multivariate statistics were applied. Non-supervised PCA suggested the existence of a different plasma amino acid profile within animal species (Fig. 3a), capable to explain 71.6% of sample variability. A hierarchical clustering of the samples represented by a heat map

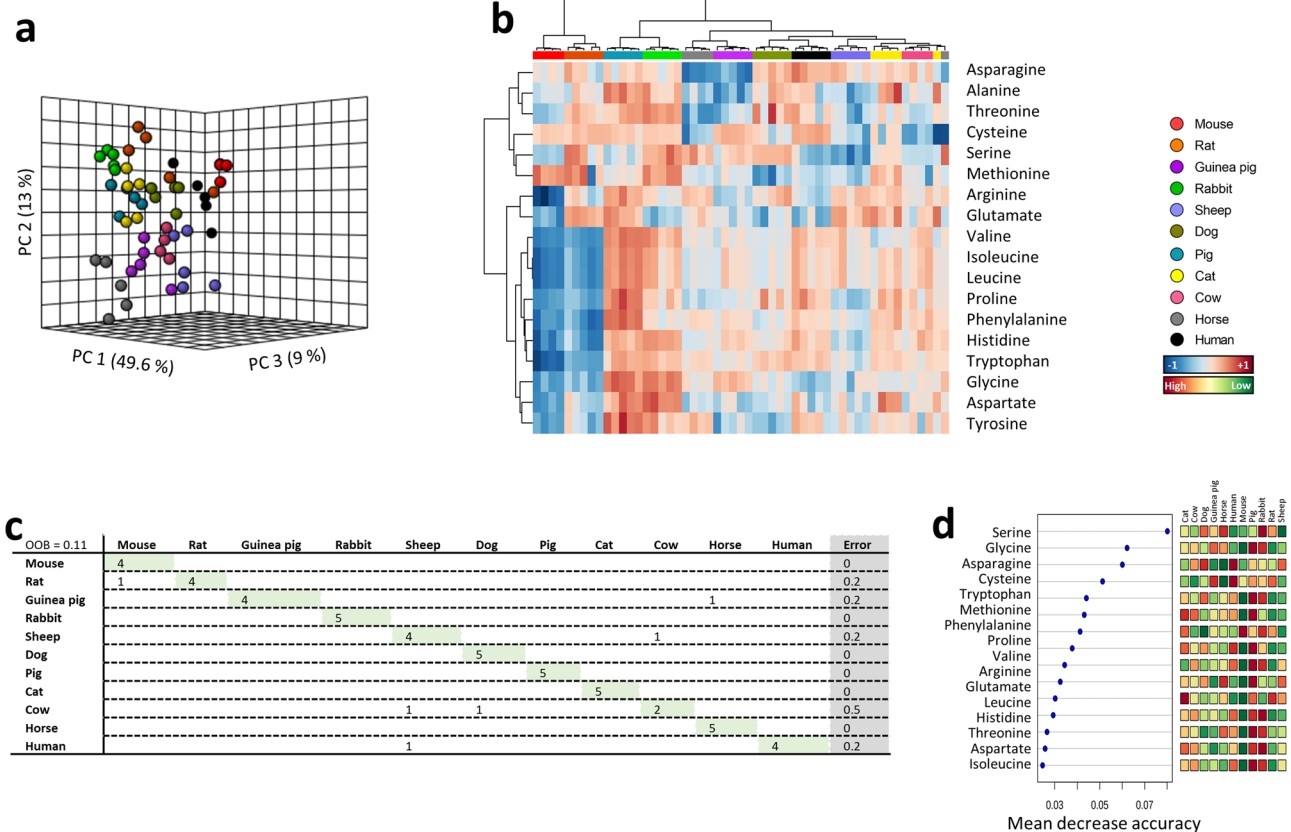

**Fig. 3 Multivariate statistics reveals a species-specific amino acids plasma profile. a** Principal component analyses (PCA) representation of amino acids. X: Principal component 1 (PC1); Y: Principal component 2 (PC2); Z: Principal component 3 (PC3). **b** Hierarchical clustering of individual animal samples according to metabolite abundance. **c** RF classification algorithm. **d** VIP scores for RF. Data values are obtained from four (mice and cows), five (rats, rabbits, guinea pigs, sheep, dogs, pigs, cats and horses) or six (humans) specimens.

confirmed the existence of a species-specific plasma amino acid profile (Fig. 3b). Accordingly, RF classification algorithm revealed a species overall classification error of 0.11, being cow the specimens with the highest classification error (50% of cow specimens were not properly classified according to its amino acids plasma profile) (Fig. 3c), being serine the metabolite with the highest contribution to classification accuracy (Fig. 3d). These data suggest that RF is a good model to identify specimens of specific mammalian species according to its methionine metabolism plasma profile.

The specific changes in plasma amino acid content across animal longevity were also evaluated (Fig. 4, Supplementary Tables 2 and 3). Among the 16 additionally detected amino acid (methionine and cysteine amino acids are not included), we have found a global increase of non-polar amino acids plasma levels in long-lived animals, such as isoleucine, leucine, phenylalanine, proline, tryptophan and valine (Fig. 4a). Plasma content of specific polar amino acids, such as asparagine, tyrosine and histidine, was also increased in long-lived animals, unlike serine and glutamate, which were decreased (Fig. 4b–d).

**Succinate and malate plasma content are decreased in long-lived animals.** Considering that amino acids can be metabolised into tricarboxylic acid (TCA) cycle intermediates, we have analysed the plasma changes for TCA cycle metabolites. Specifically, we had been able to unambiguously detect six intermediates, including pyruvate, citrate, α-ketoglutarate, succinate, fumarate and malate. In order to determine whether animal species have a specific plasma profile for TCA cycle intermediates, multivariate statistics were applied using the plasma

concentration of the mentioned metabolites. A hierarchical clustering of the samples represented by a heat map revealed the existence of a shared TCA cycle intermediates plasma profile within animal species (Fig. 5a, b).

The specific changes in plasma intermediates of TCA cycle across animal longevity were also evaluated (Fig. 5c, Supplementary Tables 2 and 3). Among the six detected metabolites, only succinate and malate were associated with animal longevity, being decreased in long-lived animals.

Amino acids and TCA cycle intermediates are bidirectionally related: the amino acids carbon skeleton can be used to synthetize TCA cycle intermediates and vice versa. Therefore, we have estimated the conversion of specific amino acids into the measured TCA intermediates across animal longevity (Supplementary Fig. 1d). In fact, long-lived animals were found to have an increased ratio of pyruvate/glycine and fumarate/aspartate. Positive correlation between fumarate and aspartate was also found (Supplementary Fig. 1e).

**Methionine-derived lipid intermediates in long-lived animals.** Methionine metabolism participates in the biosynthesis of lipid intermediates, such as choline and carnitine (Supplementary Tables 2 and 3). In long-lived animals, plasma carnitine was increased (Fig. 6a, b), whereas choline decreased (Fig. 6c).

**Pyridoxamine and succinate correlate with longevity after controlling for phylogenetic relationships.** Animal species are evolutionarily related, and closely related species often have similar traits due to inheritance from a common ancestor. Most of statistical analysis, such as linear regression, assume the

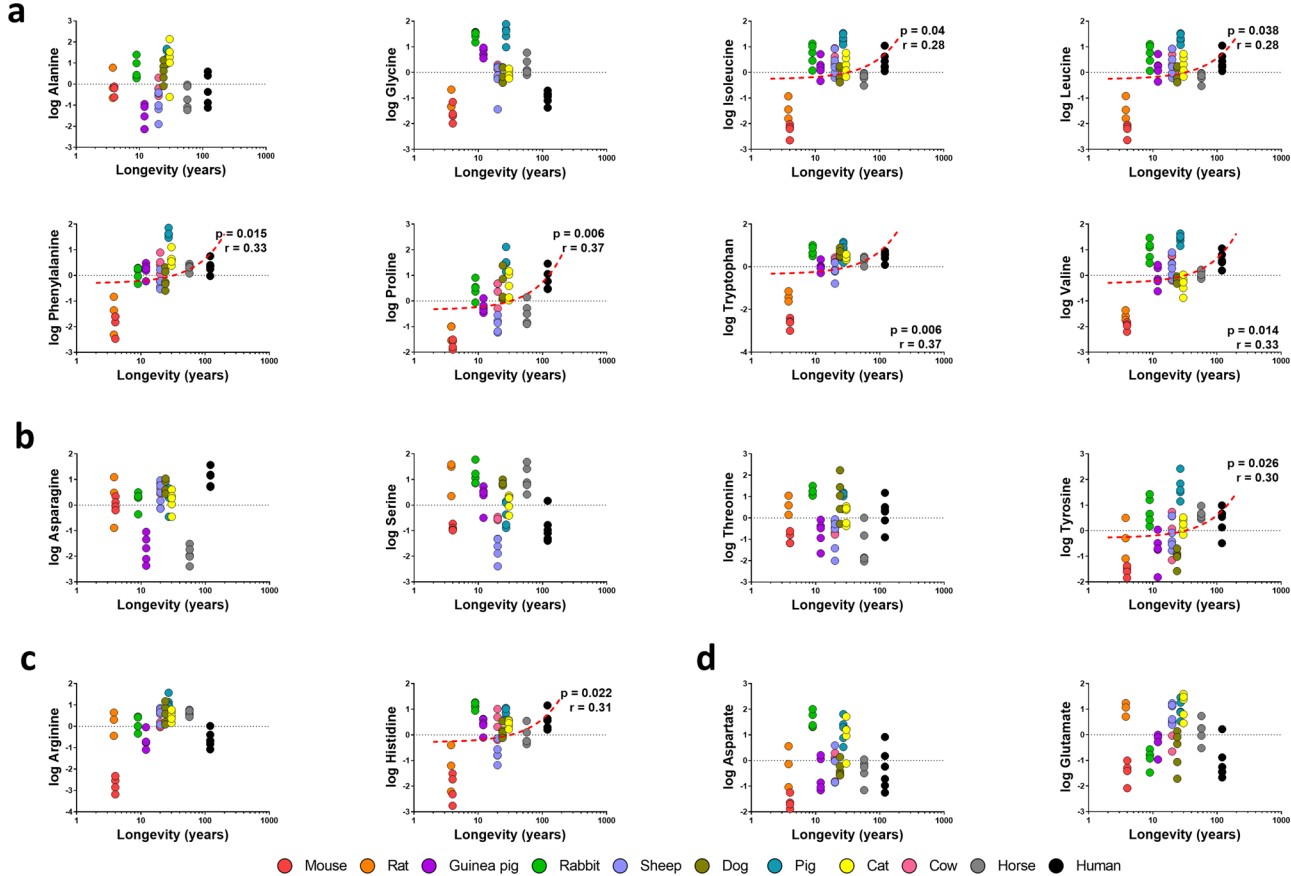

**Fig. 4 Individual plasma concentration of the detected amino acids. a** Non-polar amino acids. **b** Polar uncharged amino acids. **c** Positively charged amino acids. **d** Negatively charged amino acids. Pearson correlation between plasma metabolites and animal longevity was performed **A**–**D**. Minimum signification level was set at $p < 0.05$. All metabolites were log-transformed in order to accomplish the assumptions of normality. Data values are obtained from four (mice and cows), five (rats, rabbits, guinea pigs, sheep, dogs, pigs, cats and horses) or six (humans) specimens.

independence of the data, which might not be accomplished from data obtained from these close-related species. In order to find associations between longevity and plasma methionine metabolites, amino acids and TCA cycle intermediates, we have applied a phylogenetic comparative method, such as phylogenetically generalised least squares (PGLS) regression. A phylogenetic tree allowing to evolutionary relate the species in our study was inferred and is constructed in Fig. 7a.

First of all, under the assumption of a Brownian motion model of evolution (a branching, random walk of trait values from an ancestral value at the root to the tips of the tree[19]), we have estimated the Pagels λ. It allows to measure phylogenetic signal and indicates the relative extent to which a traits' correlation among close relatives match a Brownian motion model of trait evolution. Pagels λ range from 0 to 1, where λ = 1 indicate that trait similarities between species are influenced by phylogenetic relationships; λ = 0 indicate that trait similarities between species are independent of phylogenetic relationships; and $0 < λ < 1$ indicate different levels of phylogenetic signal. According to the estimated λ value, we have classified the measured traits according to its association degree with phylogenetic relationships (Supplementary Table 3): (1) independent (λ = 0), representing 60% of the analysed metabolites, including all the TCA cycle intermediates; (2) low dependence (λ < 0.6): methionine, taurine, pyridoxamine, glutamate and phenylalanine; (3) mild dependence (λ > 0.6): histidine, proline and tryptophan; and strong dependence (λ = 1): cystathionine, homocysteine, aspartate, tyrosine, choline. Finally, we have applied a PGLS regression, which revealed that only pyridoxamine ($p = 0.045$, $r = −0.66$) and

succinate ($p = 0.044$, $r = −0.69$) (Fig. 7b, Supplementary Table 3) plasma content were positively correlated with animal longevity after controlling for phylogenetic relationships.

## Discussion

Blood bathes cells and tissues, and serves as body's major metabolite carrier[20]. Cells release molecules to bloodstream as a consequence of their active metabolism, which will be either excreted or transported, and up taken by cells located in different tissues and organs, according to their specific metabolic needs[21]. Therefore, plasma (blood's liquid fraction) metabolome is modulated by cells and tissues, and its composition represents a global metabolic state from a living organism. Hence, plasma metabolome is the result of the metabolites that diffuse to bloodstream from each tissue as a consequence of its specific metabolism.

The achievement of a long-living phenotype requires the modulation of intracellular pathways which, in turn, regulate intracellular processes[14,22,23] that converge in synthetizing resistant intracellular structures, and limiting endogenous damage production[12]. Consequently, a globally optimized metabolic state is achieved and expressed as specific transcriptomic[24–26,19,27,28,29], proteomic[30,31], lipidomic[32–34,35] and metabolomic[36,37,13,38] profile for each tissue[38,39,40]. In the present study, we have identified the methionine-related metabolites as clue molecules defining a species-specific plasma metabolic profile. Furthermore, our model suggests that using these metabolites it is possible to identify the species of a mammalian specimen, being plasma cystathionine and

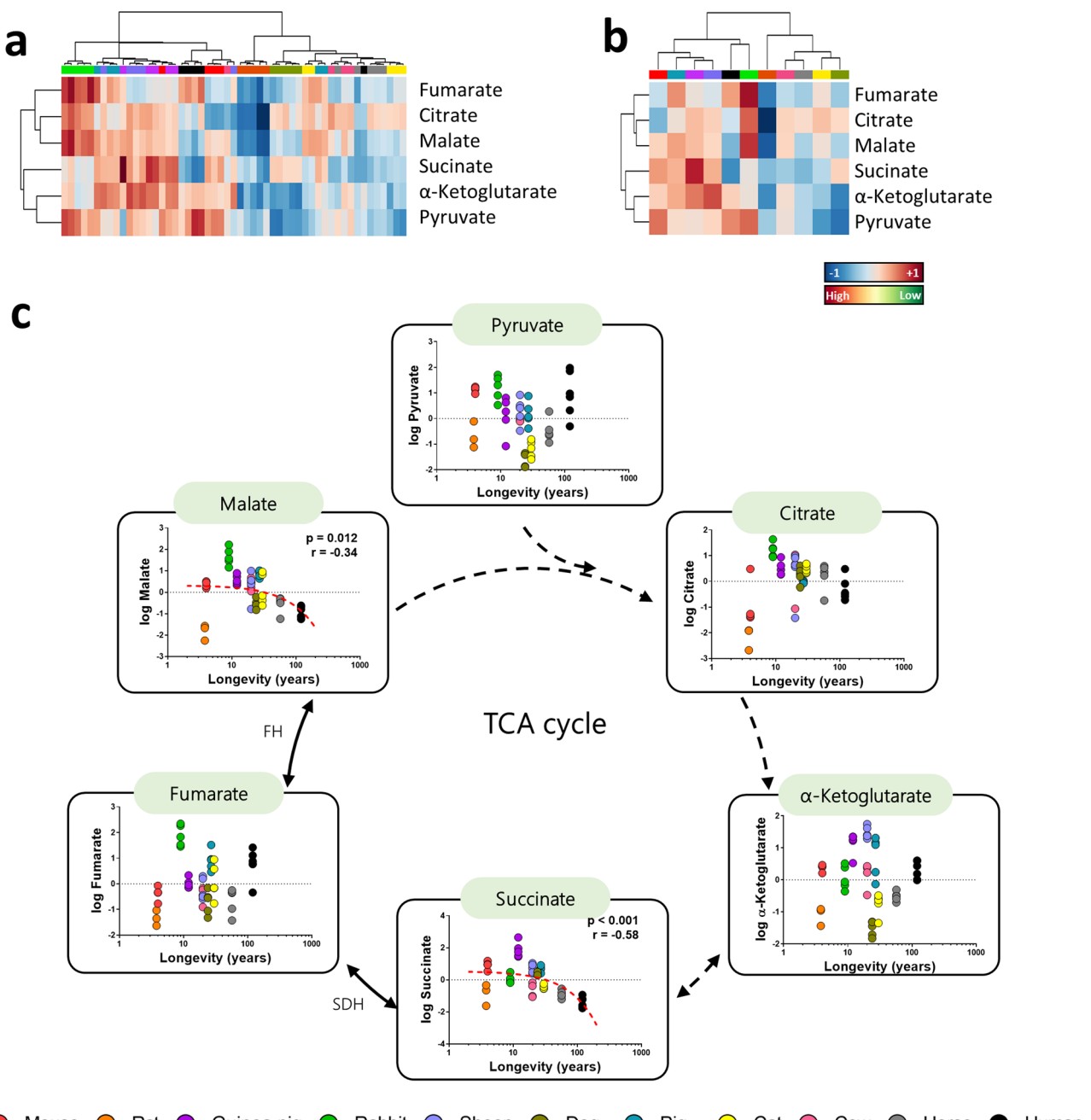

**Fig. 5 Multivariate statistics reveals a shared TCA cycle intermediates plasma profile. a** Hierarchical clustering of individual animal samples according to metabolite abundance. **b** Hierarchical clustering animal species according to average metabolite abundance. **c** Individual plasma correlation of the metabolites with animal longevity involved TCA cycle. Dashed lines refer to reactions in which more than one enzyme is involved. Pearson correlation between plasma metabolites and animal longevity was performed. Minimum signification level was set at *p* < 0.05. All metabolites were log-transformed in order to accomplish the assumptions of normality. Data values are obtained from four (mice and cows), five (rats, rabbits, guinea pigs, sheep, dogs, pigs, cats and horses) or six (humans) specimens. Enzyme codes refer to: *SDH or Cx II* succinate dehydrogenase, *FH* fumarate hydratase.

pyridoxamine, along with the amino acid serine, the highest predictors.

Reduced methionine content had been widely associated with extended longevity. Accordingly, previous studies had reported reduced DNA methionine-encoded[5], as well as decreased tissue[6,7,8] and plasma[13] content in long-lived species. Furthermore, MetR leads to an extended longevity in different experimental models[11,15,10]. However, few studies have evaluated global changes in plasma[13,41] and tissue[14] methionine metabolism intermediates associated to animal longevity. Globally, these studies suggested that species longevity is associated with reduced

sulphur-containing metabolites, and accompanied by decreased methionine plasma content[13,41]. Accordingly, our study revealed reduced steady-state plasma levels of methionine and cystathionine in long-lived mammals.

Transsulfuration, which starts with the metabolization of homocysteine into cystathionine via cystathionine-β-synthase, has been widely associated with animal longevity. Specifically, transsulfuration is enhanced in long-lived flies[17] and mice[42], in comparison to their non-longevous counterparts. The longevity effects of transsulfuration have been attributed to the generation of molecules with antioxidant properties, such as hydrogen

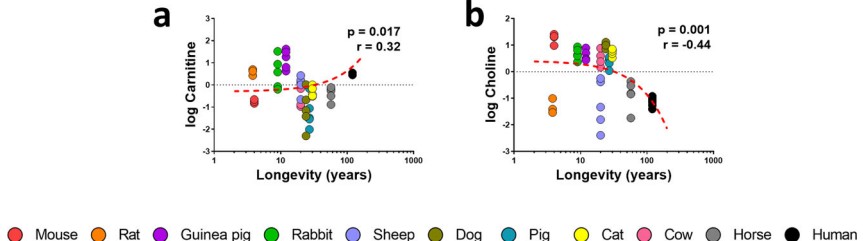

**Fig. 6 Methionine-derived lipid intermediates and its correlation with animal longevity.** Individual plasma concentration of lipid intermediates carnitine (**a**) and choline (**b**). Pearson correlation between plasma metabolites and animal longevity was performed. Minimum signification level was set at $p < 0.05$. All metabolites were log-transformed in order to accomplish the assumptions of normality. Data values are obtained from four (mice and cows), five (rats, rabbits, guinea pigs, sheep, dogs, pigs, cats and horses) or six (humans) specimens.

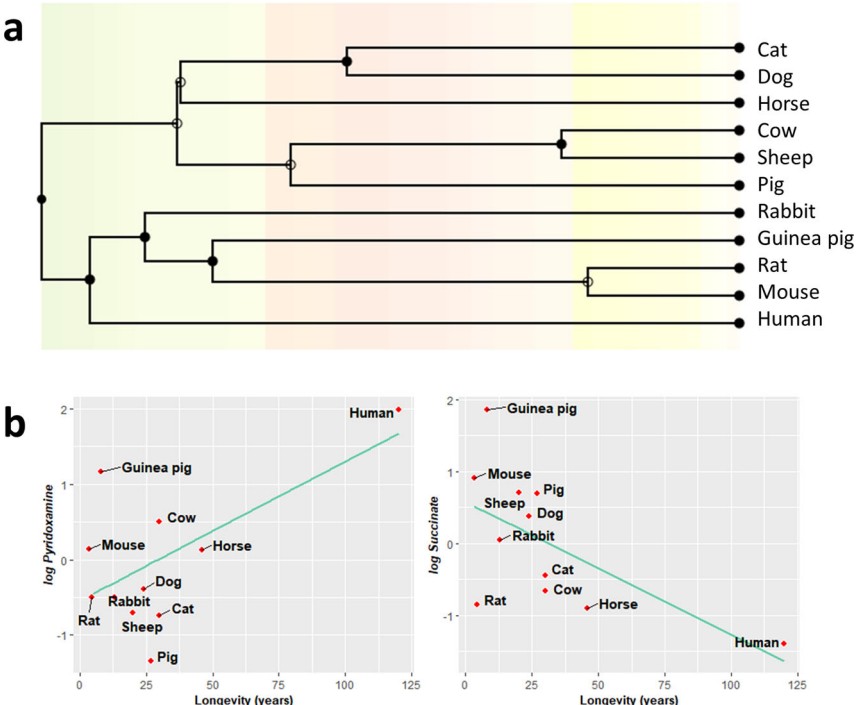

**Fig. 7 Plasma metabolites changes across mammalian longevity after correcting for phylogenetic relationships. a** Phylogenetic tree with a timescale of million years ago. **b** Phylogenetic generalized least squares (PGLS) regression between longevity and plasma content of pyridoxamine and succinate. Minimum signification level was set at $p < 0.05$. All metabolites were log-transformed in order to accomplish the assumptions of normality. Data values are obtained from four (mice and cows), five (rats, rabbits, guinea pigs, sheep, dogs, pigs, cats and horses) or six (humans) specimens.

sulfide ($H_2S$)[43–45,46], glutathione and taurine[47,48]. However, our study reveals that plasma transsulfuration intermediates or products remain unchanged (homocysteine and cysteine) across species, and even decreased (cystathionine, glutathione and taurine) in long-lived mammalian species. Under the presumption that "less is more" in long-lived species, we suggest that the steady-state levels of these intermediates are not increased since long-lived species produce less intracellular damage and are structurally built upon more resistant molecules. Therefore, these long-lived species do not need to synthetize a vast amount of antioxidant and can re-direct the energy saved to maintain other vital cellular functions. Altogether, these data support the existence of basal metabolic differences that are associated with interspecies longevity, such as decreased methionine and cystathionine plasma levels.

Proteins are essential structural components, but also the effectors of intracellular functions. Long-lived species have evolved by optimising protein composition to ensure proper protein structure, function and turnover. Synthesising more resistant proteins[49] and enhancing autophagy[50,51,52] to efficiently clear altered and non-functional proteins results in a negative correlation between global protein turnover and species longevity[53]. Accordingly, our data reveal a globally increased plasma free amino acid pool in long-lived species. Furthermore, the decreased serine plasma content, which can provide one-carbon units to regenerate methionine from homocysteine under methionine depleted conditions[54], as we reported in the present study, supports the role of methionine metabolism as determinants of animal species and longevity.

Mitochondrion structural and functional adaptations associated with longevity had been widely described, and include decreased membrane unsaturation[12], lower permeability[23,55] and modulation of mitochondrial dynamics[56]. Nonetheless, quantitative and qualitative complex I (Cx I) adaptations, such as reduced total content[12], and Cx I specific subunits[23] that support its assembly[57], resulting in reduced reactive oxygen species[58–60,57,11,12], constitutes a hallmark of longevity.

Electron transport chain is feed from electron donors derived from the TCA cycle, such as $NADH_2$ and $FADH_2$, which transfer electrons to Cx I and Cx II, respectively. As expected, TCA cycle

modulations have also been associated with animal longevity, and include the upregulation of TCA cycle genes[61,62,27,63,64], which might result in an optimised energetic metabolism. In contrast with these data, our results revealed reduced basal content of succinate and malate in plasma from long-lived species, supporting the existence of metabolic adaptations associated to species longevity, different from those specific individual adaptations associated to specimen longevity. Succinate is the substrate of Cx II, and the only TCA cycle intermediate known to trigger organismal functions regulating the immune system[65]. We suggest that since long-lived species have an optimised metabolism with more efficient Cx I, they rely less in succinate and Cx II activity to enter electrons and generate ATP, therefore, they have reduced steady levels of succinate compared to short-lived ones. Accordingly, increased plasma levels of malate in aged individuals are associated with higher cardiovascular risk [36].

Methionine metabolism and TCA cycle are connected with lipid metabolism through choline and carnitine, respectively. Choline is synthetized from betaine, and is the biosynthetic precursor of phosphocholines (PC), which can be metabolised into lysoPC or catabolised into phosphoserine by exchanging choline by serine groups[66]. Previous studies in long-lived species had already reported the existence of a specific plasma PC profile in centenarians[67,35], as well as a reduction of PC content in centenarians[67] and the exceptionally long-living naked-mole rat (NMR)[41]. Our results revealed reduced plasma choline in long-lived species, which might result in a reduced PC biosynthesis, although its reduced plasma content as a consequence of increased PC biosynthesis cannot be discarded.

Carnitine acts as a shuttle essential to transport fatty acids from the cytosol to the mitochondrial matrix, where β-oxidation occurs[68]. When combined with fatty acids, acyl-carnitines are formed. Previous studies revealed that carnitine plasma content is reduced in NMR compared to mice[41]. Our data revealed higher plasma carnitine content in long-lived species, which might suggest an improved β-oxidation. Accordingly, previous proteomic results revealed a shift in mitochondrial metabolism from mitochondrial respiration to fatty acid metabolism for energy production in NMRs in comparison to short-lived guinea pigs[30]. Supporting our results, recent studies found an association between reduced serum carnitine and human frailty[69]. Furthermore, carnitine intake has been demonstrated to have neuroprotective effects [70,68].

Comparative studies across species with different lifespan are a powerful source of information to identify mechanisms linked to extended longevity[39,38,14,23]. However, due to species evolution, interspecies studies have to deal with the problem of non-independence of the data due to evolutionary relationships[71]. Therefore, it is important to elucidate whether a specific trait correlates with longevity differences, or alternatively, these differences arise because of the data similarity, by applying statistical phylogenetic comparative methods, such as PGLS[32,72–74,14,23,28,75]. In the present study, PGLS revealed that pyridoxamine and succinate plasma content changes associated with longevity were not due interspecies relationships. Therefore, these results support the involvement of sulphur-containing metabolites biosynthesis and energetic metabolism in determining species longevity.

When performing interspecies studies, it has been a matter of controversy whether to correct or not the observed correlations for species body size and/or metabolic rate[76,77]. Although it has been suggested that there is a positive correlation between longevity and body size, up to date, it has not been attributed a physiological mechanism to body size that contributes to the ageing process. Besides, there are several species that scape this rule of thumb, such as humans, and live longer than what is expected for their body size[78]. Accordingly, many exceptions exist to the Pearl's rate of living theory of aging[79] that suggests the existence of an association between species-specific metabolic rate and longevity. Hence, we believe that correcting our data for a correlation that is not shared for all living species, or at least for one of the species included in our analysis, would lead to misleading results. Besides, as stated by Barja[76], correcting physiological associations using mathematical concepts would erase biologically meaningful information regarding species longevity.

Globally, the obtained results reveal the existence of a species-specific plasma methionine metabolic profile. Furthermore, this metabolic profile is an optimised feature of species longevity, characterised by (1) reduced plasma levels of sulphur-containing metabolites methionine and cystathionine and TCA cycle intermediates succinate and malate; (2) increased amino acids; (3) reduced choline; and (4) increased carnitine in long-lived species.

It is important to remark that although plasma composition reflects organismal metabolic status, we cannot disregard the possibility of the existence of controlled mechanisms regulating plasma composition, such as metabolite transports, which might differ across tissue and organs. Furthermore, intracellular function of the described metabolites has already been described, but a lack of knowledge is found regarding plasmatic function of plasma metabolites. Therefore, more studies focused on elucidating the function of the plasmatic profiles associated to longevity is needed.

Apart from lack-of-knowledge limitations, our work also has some technical limitations. First, the fact that although metabolome offers a wide amount of information, we cannot rule out the fact that plasma metabolome is highly affected by internal factors, such as tissue metabolism, or mixture of plasma with other biological fluids during animal sacrifice; and external factors, such as diet or circadian rhythms[80,81]. Nevertheless, we believe that these changes might be a reflection of individual or specimen-specific plasma metabolome modulations within a specie, and that the obtained plasma profile in our work reveals species-specific organismal metabolic adaptations associated to mammalian longevity. Besides, another limitation of our work is that we have only included a set of eleven mammalian species, which represents a minor percentage of living species. Therefore, it would be interesting to increase the number of analysed species, and also include species with similar body size but different longevities, such as birds and rodents.

The present work describes a species-specific longevity profile, which is the result of a whole organismal metabolic reorganisation that exert a "big effect" on longevity, but needs to be differentiated from other metabolic adaptations exerting a "small effect" on longevity, as suggested previously[22]. The later, represent the individual metabolic adaptations resulting from genetic modulations or nutritional interventions that determine whether a specimen from a specific specie becomes long-lived on not. Most of the efforts have been directed to study the longevity metabolomic determinants of long-lived specimens[13,82,41], vertebrate[83,42] and invertebrate long-lived mutants[84,17], or its modulation by nutritional interventions increasing longevity[85,86,87]. The results obtained from these studies are of big value since allow to describe strategies that would be feasible to apply to humans in order to improve our health span. However, these results only represent the "small effect" determinants of longevity, and might resemble, such as decreased Cx I[23,57,6] and/or reduced intracellular damage[6,88,8], or differ, such as plasma metabolic profile[82,83,13,69,41,87], from those obtained when comparing species of different longevities, as described previously. In the recent years, the number of interspecies studies have increased, since they provide the basal metabolic status of each species and to compare it with other species, which is

essential to unravel the mechanisms underlying an improved longevity and to properly design strategies to improve human health span.

## Methods

**Chemicals**. Unless otherwise specified, all reagents were from Sigma-Aldrich, and of the highest purity available.

**Samples**. Mammalian species included in the study were male adult specimens with an age representing their 15–30% of their longevity. The recorded values for longevity (in years) were: mouse (*Mus musculus*, n = 4), 4; rat (*Rattus norvegicus*, n = 5), 3.8; rabbit (*Oryctolagus cuniculus*, n = 5), 9; guinea pig (*Cavia porcellus*, n = 5), 12; sheep (*Ovis aries*, n = 5), 20; cow (*Bos taurus*, n = 4), 20; dog (*Canis lupus*, n = 5), 24; pig (*Sus scrofa*, n = 5), 27; cat (*Felis catus*, n = 5), 30; horse (*Equus caballus*, n = 5), 57; and human (*Homo sapiens*, n = 6), 120. Longevity values chosen for dog and sheep samples are approximated, and do not correspond to a specific breed. Rodent and rabbit specimens were obtained from rodent husbandries and blood samples were obtained after sacrifice by decapitation. For humans, blood samples were acquired from healthy adult individuals by venipuncture. The animal care protocols were approved by the Animal Experimentation Ethics Committee of the University of Lleida, and all relevant ethical regulations were complied. Human protocols were approved by the Committee for Ethics in Clinical Research of the Hospital Universitari Arnau de Vilanova, in accordance with the Declaration of Helsinki. All subjects were fully informed of the aims and scope of the research and signed an informed consent. Blood samples from rodent and rabbit specimens, and subjects, were obtained in the morning after fasting (8–12 h). Blood samples were centrifuged to separate plasma fraction, which was immediately frozen in liquid nitrogen and stored at −80 °C before 4 h until analyses. For dog, cat, sheep, pig, cow, and horse specimens, blood samples were acquired by venipuncture in the morning after fasting (8–12 h), and after centrifugation, plasma samples were obtained and preserved at −80 °C. Blood acquisition and processing to obtain plasma was performed by Charles Rivers Laboratories France Iffa-Credo (France).

**Sample processing**. Plasma metabolites extraction was performed based on the methodology previously described (Method 1[89]). Briefly, 10 μL of plasma were added to 30 μL of cold methanol containing 1 μg/mL of Phe-$^{13}$C as internal standard and 1 μM BHT as antioxidant. Then, samples were incubated at room temperature for 15 min and centrifuged at 12,000 × g for 3 min. Finally, the supernatant was filtrated through a 0.22-μm organic diameter filter (CLS8169, Sigma, Madrid, Spain) and transferred to Agilent (Barcelona, Spain) vials with glass inserts for further analysis.

Sulphur-containing metabolites were extracted on the bases of the methodology previously described (Method 2[90]). Briefly, 2 μL of 5% DTT diluted in methanol (m/v) were added to 10 μL of plasma. The resulting solution was vortexed for 1 min and allowed to stand at room temperature for 10 min. For protein precipitation, 40 μL of acetonitrile containing 0.1% formic acid (v/v), 0.05% trifluoroacetic acid (v/v) and 1 μg/mL of Phe-$^{13}$C as internal standard was added to the sample, and the solution was vortexed for 2 min. Then, samples were incubated at room temperature for 15 min and centrifuged at 12,000 × g for 3 min. Finally, the supernatant was filtrated through a 0.22-μm organic diameter filter (CLS8169, Sigma, Madrid, Spain) and transferred to Agilent (Barcelona, Spain) vials with glass inserts for further analysis.

**Analysis conditions**. The individual conditions for the detection and quantification of plasma metabolites are listed in Supplementary Table 1. For non-sulphur-containing metabolites, 2 μL of extracted sample was injected based on the method described (Method 1[89]). Chromatographic separation was achieved on a reversed-phase column (Zorbax SB-Aq 2.1 × 50 mm, 1.8 μm particle size, Agilent Technologies, Barcelona, Spain) equipped with a pre-column (Zorba-SB-C8 Rapid Resolution Cartridge, 2.1 × 30 mm, 3.5 μm particle size, Agilent Technologies, Barcelona, Spain) with a column temperature of 60 °C. The flow rate was 0.6 mL/min during 19 min. Solvent A was composed of water containing 0.2 % acetic acid (v/v) and solvent B was composed of methanol containing 0.2 % acetic acid (v/v). The gradient started at 2 % of solvent B and increased to 98 % B in 13 min and held for 6 min. Post-time was established in 5 min. Electrospray ionization was performed in both positive and negative ion mode (depending on the target metabolite) using N$_2$ at a pressure of 50 psi for the nebulizer with a flow of 12 L/min and a temperature of 325 °C, respectively.

For sulphur-containing metabolites, 10 μL of extracted sample was injected based on the method described (Method 2[90]). Chromatographic separation was achieved on a reversed-phase Supelcosil LC-CN column (Supelco 4.6 × 250 mm, 5 μm particle size, Sigma, Madrid, Spain) with a column temperature of 30 °C. The flow rate was maintained at 0.5 mL/min during 10 min using a mobile phase of 10:90 acetonitrile/water with 0.1% formic acid (v/v). Electrospray ionization was performed in both positive and negative ion mode (depending on the target metabolite) using N$_2$ at a pressure of 50 psi for the nebulizer with a flow of 12 L/min and a temperature of 325 °C, respectively.

Data were collected using the MassHunter Data Analysis Software (Agilent Technologies, CA, USA). Samples were decoded and randomized before injection. Metabolite extraction quality controls (plasma samples with internal Phe-$^{13}$C) were injected every ten samples. Peak determination and peak area integration were carried out with MassHunter Quantitative Analyses (Agilent Technologies, CA, USA).

**Equipment**. The analysis was performed through liquid chromatography coupled to a hybrid mass spectrometer with electrospray ionization and a triple quadrupole mass analyser. The liquid chromatography system was an ultra-performance liquid chromatography model 1290 coupled to LC-ESI-QqQ-MS/MS model 6420 both from Agilent Technologies (Barcelona, Spain).

**Statistics and reproducibility**. Prior to statistical analyses, data were pre-treated (auto-scaled and log-transformed). Multivariate statistics was performed using Metaboanalyst software[91], and include PCA, hierarchical clustering analysis represented by a heat map, and RF used as a classification algorithm of animal species according to its plasma metabolome. Univariate statistics (Pearson correlation, Pearson correlation matrix, linear models and PGLS regression) were performed using RStudio (v1.1.453). Correlation functions were included in the packages *Hmisc*[92] and *corrplot*[93], and plotted with *ggplot2*[94]. Linear regression was plotted using GrapPhad Prism (v8.0.1). PGLS regression functions were included in the package *caper*[95], and used to analyse the correlation of individual metabolites with animal longevity after controlling for phylogenetic relationships, defined by a phylogeny. The phylogenetic tree was constructed using taxa names as described previously[96]. To ensure data reproducibility, 4–6 independent specimens for each species was included in the analyses.

**Reporting summary**. Further information on research design is available in the Nature Research Reporting Summary linked to this article.

## Data availability

All data generated or analysed during this study are included in this published article and its supplementary information files (Supplementary Data 1)

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

## Acknowledgements

This work was supported by the Spanish Ministry of Science, Innovation and Universities (RTI2018-099200-B-I00), and the Generalitat of Catalonia (Agency for Management of University and Research Grants (2017SGR696) and Department of Health (SLT002/16/00250)) to R.P. This study has been co-financed by FEDER funds from the European Union ("A way to build Europe"). IRBLleida is a CERCA Programme/Generalitat of Catalonia. M.J. is a 'Serra Hunter' Fellow. N.M.M. received a predoctoral fellowship from the Generalitat of Catalonia (AGAUR, ref 2018FI_B2_00104).

## Author contributions

R.P designed the study. N.M.M., M.J and R.B. performed experimental work. N.M.M, M.J. and R.P. analysed the data. R.P. supervised the design and data interpretation. The manuscript was written by N.M.M, M.J. and R.P. and edited by R.P. All authors discussed the results and commented on the manuscript.

## Competing interests

The authors declare no competing interests.
