## [Peer Review File · Communications Biology]

Reviewers' comments:

Reviewer #1 (Remarks to the Author):

This study used a comparative metabolomic approach to examine plasma levels of specific metabolites linked primarily to methionine metabolism and reported a species-specific profile that correlated with reported lifespans. While I believe there is a tremendous amount one can learn regarding modulators of lifespan using such a comparative approach, my enthusiasm is dampened by the lack of experimental rigor and a suite of alternative explanations for these observations.

a) The maximum lifespan of the species used in this study does not concur with those on the AnAge website (<https://genomics.senescence.info/species/query.php?search>) with in some cases pronounced differences in maximum lifespan. E.g., Mouse 4y versus 3.5y; Rat 3.8y versus 4.5y; guinea pig 12 y versus 8 y; rabbit 9y versus 13y; cow 20y versus 30y, horse 57y versus 46y and instead of including domestic pig (which I presume they used) they provide the longevity for a wild boar. These data would need to be reanalyzed with these referenced and more reliable maximum lifespan reports. It is well documented that the size and breed of dog influences its maximum lifespan with small dogs living close to the maximum reported here and large dogs living approximately 1/3 as long. Which breeds of dogs were used in these analyses? Most of these longevities also correlate with species body size and/or metabolic rate, yet no mention is made here if the plasma metabolites were analyzed with respect to body size.

b) Blood was collected by different methods- decapitation for rodents and no information as to how this was collected for domesticated species and by venipuncture for humans. Blood collected by decapitation mixes blood with lymph and interstitial fluids and will alter the concentration of plasma metabolites. Were the blood samples collected at the same time of day or at a set number of hours after feeding? Both circadian rhythms and time since last meal will affect plasma content and in particular that of amino acids.

c) While I appreciate that the choice of species used is based upon that which is readily available, this diverse set of species also have different diets e.g., herbivore/omnivore and carnivore and this too will influence their plasma metabolome. It would have been better if several rodents of different longevity were included; in their phylogenetic tree the authors include gerbil but don't provide any data for this common species available from most rodent suppliers and if multiple representatives with different longevities from the various clades were used in these analyses.

Reviewer #2 (Remarks to the Author):

Intake and metabolism of methionine have been linked to longevity in multiple species. The purpose of this study was to define the methionine metabolic phenotype in long-lived species via measurement of 35 plasma metabolites in 11 distinct mammalian species representing a broad range of lifespans, ranging from those of mice to humans. The results were interesting, and with some alterations, the manuscript could be improved:

1. In the first paragraph of the Discussion, the authors note that the plasma metabolome "represents a global metabolic state from a living organism." However, it is also important to note that not every tissue will necessarily reflect or correlate with what is observed in the circulation.
2. The authors posit that glutathione levels are lower in long-lived species because those species naturally require less antioxidant protection. Please elaborate further.
3. The authors also posit that malate and succinate, rather than any other TCA cycle intermediates, are lower in long-lived species due to metabolic alterations. Please elaborate further on these ideas.
4. In Fig. 6B, the relationships between longevity and (A) log Pyridoxamine and (B) log Succinate appear to be disproportionately affected by the data point from humans. What does the relationship look like without that human data point?
5. In Fig. 7, the shapes of the curves between longevity and (A) pyruvate/gly and (B)

fumarate/asp seem to be dependent mostly on data from mice, rats, and humans rather than any other species. If those point are removed, is the shape of the curve dramatically altered?

Letter Response to Reviewers
Ref. COMMSBIO-20-2320

Reviewer #1 (Remarks to the Author):

This study used a comparative metabolomic approach to examine plasma levels of specific metabolites linked primarily to methionine metabolism and reported a species-specific profile that correlated with reported lifespans. While I believe there is a tremendous amount one can learn regarding modulators of lifespan using such a comparative approach, my enthusiasm is dampened by the lack of experimental rigor and a suite of alternative explanations for these observations.

R/ We appreciate the reviewer's comments

- a) The maximum lifespan of the species used in this study does not concur with those on the AnAge website (<https://genomics.senescence.info/species/query.php?search>) with in some cases pronounced differences in maximum lifespan. E.g., Mouse 4y versus 3.5y; Rat 3.8y versus 4.5y; guinea pig 12 y versus 8 y; rabbit 9y versus 13y; cow 20y versus 30y, horse 57y versus 46y and instead of including domestic pig (which I presume they used) they provide the longevity for a wild boar. These data would need to be reanalysed with these referenced and more reliable maximum lifespan reports. It is well documented that the size and breed of dog influences its maximum lifespan with small dogs living close to the maximum reported here and large dogs living approximately 1/3 as long. Which breeds of dogs were used in these analyses? Most of these longevities also correlate with species body size and/or metabolic rate, yet no mention is made here if the plasma metabolites were analysed with respect to body size.

R/ According to the reviewer's comments, we have reanalysed the data using the corrected longevities for mouse, rat, guinea pig, rabbit, cow and horse. Please see text highlighted in yellow in page 5 and Figures 2, 4-6 5, and Supplementary figure 1.

We have opted to maintain pig and dog longevity values for several reasons. We are aware of the different longevities reported for different species within a genus. However, we believe that this is a limitation that inter-species species studies assume. Accordingly, other inter-species studies have already used the same longevity values for pigs and dogs independently of the breed (doi: 10.1111/j.1474-9726.2011.00718.x). Besides, other studies aimed to elucidate longevity-associated mechanisms have extrapolated *Sus scrofa* longevity values to other species belonging to the genus *Sus* (doi: 10.1186/1471-2164-15-653).

When performing inter-species studies, it has been a matter of controversy whether to correct or not the observed correlations for species body size and/or metabolic rate (doi: 10.1093/gerona/glu020; 10.1111/j.1474-9726.2005.00162.x). We have discussed this issue in the discussion section of the MS. Please see text highlighted in yellow in page 17.

- b) Blood was collected by different methods- decapitation for rodents and no information as to how this was collected for domesticated species and by venipuncture for humans. Blood collected by decapitation mixes blood with lymph and interstitial fluids and will alter the concentration of plasma metabolites. Were the blood samples collected at the same time

of day or at a set number of hours after feeding? Both circadian rhythms and time since last meal will affect plasma content and in particular that of amino acids.

R/ The previous description was a misunderstanding as it was referred to another MS that we are preparing at the moment to be submitted about an interspecies study using tissue (heart) samples. Our apologies for the inconveniences. In the new version of the MS we have corrected the method sections and included the details of plasma sample acquisition for each species. Please see text highlighted in yellow in page 5.

We are aware of the technical limitations of our work, due to sample collection and metabolome modulation by external factors (circadian rhythm, sacrifice methods, animal diet, among others). We have included this limitation in the discussion section of the MS. Please see text highlighted in yellow in page 17-18.

To the best of our knowledge, studies comparing plasma, lymph and interstitial fluid (IF) metabolome are limited, and mostly performed under pathological conditions. Only two studies evaluated lymph metabolome (doi: 10.2450/2016.0208-15; 10.1007/s12127-012-0102-4). In rodents, 1200-34000 metabolites were detected in lymph, and mostly weighting 500-750 Da (doi: 10.1007/s12127-012-0102-4), which is 5-fold less than the 18557 detected and quantified metabolites in human plasma (doi: 10.1093/nar/gkx1089). The unique study comparing plasma and lymph metabolome was performed in humans after a trauma injury. In this study, it was revealed a positive and strong ($r > 0.75$) concentration between plasma and lymph content of 50% of the detected metabolites, including amino acids, methionine-related metabolites and TCA cycle intermediates. As for lymph metabolome, only two studies compared the metabolome of IF and plasma (doi: 10.3390/diagnostics10110936; 10.1152/ajpendo.00156.2018), one of them to identify hypertension biomarkers in oncological patients (doi: 10.3390/diagnostics10110936). In rodents and humans, 43% of the detected metabolites were shared in plasma and muscle IF, and only 20% of those changed in response to physical activity (doi: 10.1152/ajpendo.00156.2018). Therefore, although we cannot discard the fact that plasma metabolome might be affected by the presence of lymph and interstitial fluids after animal decapitation, we believe that present evidence doesn't point to a high effect of the selected sacrifice method on plasma metabolome composition due to its temporary (no more than a few seconds) mixture with other biological fluids.

c) While I appreciate that the choice of species used is based upon that which is readily available, this diverse set of species also have different diets e.g., herbivore/omnivore and carnivore and this too will influence their plasma metabolome. It would have been better if several rodents of different longevity were included; in their phylogenetic tree the authors include gerbil but don't provide any data for this common species available from most rodent suppliers and if multiple representatives with different longevities from the various clades were used in these analyses.

R/ As stated previously, the previous figure 7 was a misunderstanding as it was referred to another MS that we are preparing at the moment to be submitted about an interspecies study using tissue (heart) samples and other mammalian species. Our apologies for the inconveniences. In the new version of the MS we have corrected the phylogenetic tree removing gerbil (see Figure 7). Besides, we have discussed the possible effect of diet and the

possibility to increase the number of species included in the analyses in the discussion section of the MS. Please see text highlighted in yellow in page 16-17.

Reviewer #2 (Remarks to the Author):

Intake and metabolism of methionine have been linked to longevity in multiple species. The purpose of this study was to define the methionine metabolic phenotype in long-lived species via measurement of 35 plasma metabolites in 11 distinct mammalian species representing a broad range of lifespans, ranging from those of mice to humans. The results were interesting, and with some alterations, the manuscript could be improved:

R/ We appreciate the reviewer's comments

1. In the first paragraph of the Discussion, the authors note that the plasma metabolome represents a global metabolic state from a living organism. However, it is also important to note that not every tissue will necessarily reflect or correlate with what is observed in the circulation.

R/ We agree with the reviewer's comment. We are not stating that plasma metabolites correlate with every tissue-specific metabolome. In fact, we suggest that plasma metabolome is the result of the metabolites that diffuse to plasma from each tissue as a consequence of tissue-specific metabolism. We have included a sentence at the end of the first paragraph in the discussion section of the MS. Please see text highlighted in yellow in page 16-17.

2. The authors posit that glutathione levels are lower in long-lived species because those species naturally require less antioxidant protection. Please elaborate further.

R/ Please see highlighted text in yellow in page 14 of the MS.

3. The authors also posit that malate and succinate, rather than any other TCA cycle intermediates, are lower in long-lived species due to metabolic alterations. Please elaborate further on these ideas.

R/ Please see highlighted text in yellow in page 15 of the MS.

4. In Fig. 6B, the relationships between longevity and (A) log Pyridoxamine and (B) log Succinate appear to be disproportionately affected by the data point from humans. What does the relationship look like without that human data point?

R/ We are aware that human data might affect correlations. However, we have evaluated the possibility to repeat the performed analyses removing human data, to improve correlation from a mathematical point of view. However, we have finally opted for maintaining human values as we believe that, from a biological and gerontological perspective, it's more valuable to be able to determine how long-lived mammals (as humans are considered to be) adapt its metabolism in comparison to short-lived mammals. Hence, we prefer to maintain the data as we present in the first version of the MS.

5. In Fig. 7, the shapes of the curves between longevity and (A) pyruvate/ gly and (B) fumarate/ asp seem to be dependent mostly on data from mice, rats, and humans rather than any other species. If those point are removed, is the shape of the curve dramatically altered?

R/ As for the previous comment, we prefer to maintain the data as we have presented in the first version of the MS as discussed in the previous comment.

REVIEWERS' COMMENTS:

Reviewer #1 (Remarks to the Author):

Plasma methionine metabolic profile is associated with longevity in mammals

The authors have addressed many of the concerns that I had previously raised. I appreciate why they have chosen to keep maximum lifespan for dogs and sheep not based upon the specific data for the breed or species and refer to others who have similarly not done so. While I'm OK to let this go, although this is only an approximation; I think the caveats of this should be mentioned in the methods. Line 436-449 seem out of place and this certainly was not what I was asking for in my review; rather I was asking if you had plotted your data as a function of body size instead of as a function of max lifespan. I would shorten this and add it to the section on limitations of the study lines 463ff

Minor typos need to be corrected.

e.g.

Lines 336-337 "Few studies have unfocused from methionine and evaluated global changes in plasma

Line 346 fly should be flies or the long lived fly

Lines 413/417 the species should be listed as plurals or with the word "the" in front of it should be NMRs or the NMR

Reviewer #1 (Remarks to the Author):

Plasma methionine metabolic profile is associated with longevity in mammals

The authors have addressed many of the concerns that I had previously raised. I appreciate why they have chosen to keep maximum lifespan for dogs and sheep not based upon the specific data for the breed or species and refer to others who have similarly not done so. While I'm OK to let this go, although this is only an approximation; I think the caveats of this should be mentioned in the methods.

R/ We have included a sentence. Please see text highlighted in green in the new version of the MS

Line 436-449 seem out of place and this certainly was not what I was asking for in my review; rather I was asking if you had plotted your data as a function of body size instead of as a function of max lifespan. I would shorten this and add it to the section on limitations of the study lines 463ff

R/ Our apologies for the misunderstanding. We had plotted our data as a function of body size and most of the metabolites were not significantly correlated with species body size. However, as we had stated in the previous version (lines 436-442), we believe that as this correlation is not shared for all living species, we prefer not to correct our data for the effect of each species body size. We prefer to maintain the lines 436-442 in the discussion section, as we do not believe that it is not a limitation, but to justify our results.

Minor typos need to be corrected.

e.g.

Lines 336-337 “;Few studies have unfocused from methionine and evaluated global changes in plasma

Line 346 fly should be flies or the long lived fly

Lines 413/417 the species should be listed as plurals or with the word “the”; in front of it should be NMRs or the NMR

R/ We have corrected the typos. Please see text highlighted in green in the new version of the MS.